# The Antifungal Activity of Cinnamon-Litsea Combined Essential Oil against Dominant Fungal Strains of Moldy Peanut Kernels

**DOI:** 10.3390/foods11111586

**Published:** 2022-05-28

**Authors:** Yijun Liu, Ruolan Wang, Lingli Zhao, Shanshan Huo, Shichang Liu, Hanxiao Zhang, Akio Tani, Haoxin Lv

**Affiliations:** 1College of Food Science and Engineering, Henan University of Technology, Zhengzhou 450001, China; lyj3500@163.com (Y.L.); wangruol@163.com (R.W.); zhaolingli_123@163.com (L.Z.); 18638003053@163.com (S.H.); liu18135662718@163.com (S.L.); smile410526@163.com (H.Z.); 2Institute of Plant Science and Resources, Okayama University, Okayama 710-0046, Japan; atani@okayama-u.ac.jp

**Keywords:** anti-mildew agents, peanut storage, cinnamon-litsea combined essential oil (CLCEO), antifungal activity, *Aspergillus flavus*

## Abstract

The antifungal activity of cinnamon (*Cinnamomum cassia* Presl), litsea [*Litsea cubeba* (Lour.) Pers.], clove (*Syzygium aromaticum* L.), thyme (*Thymus mongolicus* Ronn.) and citronella (*Cymbopogon winterianus* Jowitt) essential oils (EOs) against the dominant fungi isolated from moldy peanuts was investigated in this research. Firstly, strain YQM was isolated and identified by morphological characterization and 18S rRNA gene sequence analysis to be *Aspergillus flavus* (*A*. *flavus*). Next, antifungal effects of single or mixed EOs on strain YQM were evaluated by the inhibition zone test. The cinnamon-litsea combined essential oil (CLCEO, V_cinnamon oil_:V_litsea oil_ = 3:5) displayed the best antifungal effect on strain YQM. The chemical composition of CLCEO was identified and quantified by gas chromatograph-mass spectrometry (GC-MS), and results revealed that the major components of CLCEO were cinnamaldehyde and citral. Finally, the effect of EOs on the microstructure of strain YQM mycelia was observed under scanning electron microscope (SEM). The mycelia exposed to cinnamon essential oil (CEO) and litsea essential oil (LEO) were partly deformed and collapsed, while the mycelia treated with CLCEO were seriously damaged and the deformation phenomena such as shrinking, shriveling and sinking occurred. Therefore, CLCEO has great potential for using as anti-mildew agents during peanut storage.

## 1. Introduction

*Arachis hypogaea* Linn., commonly known as peanut or groundnut, is an important oil crop grown in most parts of the world [1]. Due to the characteristics of poor heat resistance and high hygroscopicity, peanuts easily go moldy under improper storage conditions. It has been reported that peanuts are susceptible to infestation by *Aspergillus* spp., such as *Aspergillus flavus* (*A*. *flavus*) and *A*. *parasiticus*, during the process of post-harvest storage [2]. The proliferation of *Aspergillus* spp. does not only reduce the germination rate and the protein content of peanuts, but also leads to lipid oxidation, which causes huge quality loss and spoilage [3,4]. Moreover, mycotoxins produced by *Aspergillus* spp., especially aflatoxins, contaminate peanuts and endanger human and animal health [5]. From 2015 to 2018, cases of excessive aflatoxins in peanuts accounted for 93% of violations of mycotoxins in Chinese food and food exports to the EU, which seriously restricts the development of Chinese peanut industry [6]. Therefore, it is urgent to find effective methods to inhibit the growth of molds during the peanut storage process.

For many years, synthetic fungicides, as an efficient method, have been widely used to control the growth of phytopathogenic fungi to reduce grain losses and ensure global food security [7,8]. Although synthetic fungicides are effective against a variety of molds, adverse effects on the environment and the development of resistance increased public concern [9]. Most synthetic chemicals can cause a certain accumulated toxic effect on human beings, environmental pollution, and drug resistance after long-term use, these factors lead to the restricted use of chemical synthetic antifungal agents [10,11]. Due to the negative effects of synthetic fungicides on stored grains, it is absolutely imperative to develop safe and natural anti-mildew compounds in the replacement of chemical synthetic ones [12].

Essential oil (EOs), secondary metabolites extracted from plant materials, are expected to be safe antimildew agents and have received increasing attention due to their effective antifungal activity, environmental soundness, and low toxicity [13]. Many researchers have investigated the antifungal effects of EOs against fungi commonly found in grains. Reyes-Jurado et al. [14] studied the antifungal activity of thyme EO in vapor-phase against *A*. *niger* and *Penicillium expansum* in corn tortillas, and they found thyme EO significantly slowed the growth of *A*. *niger*. Strelkova et al. [15] assessed the inhibitory activity of thyme, oregano, lemongrass, clove, and cajeput EOs against five strains of *Penicillium ochrochloron*, *Fusarium sporotrichioides*, *Fusarium solani*, *A*. *niger*, and *A*. *flavus*, and found that the growth of the five fungal strains could be completely inhibited by EOs when each of them was used at 125 µL/L, except cajeput essential oil (EO), among which the effect of thyme EO was the best. Ju et al. [16] reported that cinnamon and clove EOs could inhibit the growth of *Penicillium* spp. and *Aspergillus* spp. and they also could extend the shelf life of green bean cake by about 9–10 and 3–4 days, respectively. Anžlovar et al. [17] found that thyme EO could inhibit the growth of endophytic fungi in wheat without affecting seed germination. The results of Raveau et al. [18] also confirmed the potential of EOs (clary sage and coriander seeds EO) as post-harvest decay control products to inhibit the growth of *Zymoseptoria tritici* and *Fusarium culmorum*.

Though some progress has been made in the application of plant EOs in grain storage, there are still problems to be solved in practical application, such as the high cost of EOs. Combined EOs not only improve the antifungal effect, but also reduce the dosage of EOs, which reduces the costs and the odor of EOs [19]. Purkait et al. [20] indicated that the combined EO composed of cinnamon and clove EOs had a synergistic antifungal effect on *A*. *niger* and was recommended as a safe and natural antifungal blend in the food and pharmaceutical industries. Yuan et al. [21] reported that the combination of thymol, carvacrol, *trans*-cinnamaldehyde, eugenol and vanillin exhibited additive interactions against *Escherichia coli* O157:H7, and the reason for the stronger antimicrobial efficacy of EO compounds was due to the increase of membrane damages. Hlebová et al. [22] observed three combinations of litsea and lemongrass, clove and lemongrass, cinnamon bark and lemongrass EOs; all showed synergistic effect against *A*. *flavus*, and the mixture of EOs with synergistic antifungal properties can reduce resistance and the dosage of EOs. These previous studies have shown that combined EOs are more effective in preventing the contamination of cereals by molds than single EOs. However, the research data on the application of combined EOs in peanut storage still needs to be supplemented in future research [23,24].

Therefore, the aims of this study were to determine the antifungal efficacy of five single EOs including cinnamon (*Cinnamomum cassia* Presl) EO (CEO), litsea [*Litsea cubeba* (Lour.) Pers.] EO (LEO), clove (*Syzygium aromaticum* L.) EO (CLO), thyme (*Thymus mongolicus* Ronn.) EO (TEO) and citronella (*Cymbopogon winterianus* Jowitt) EO (CNO) and to select the combined EO with the best inhibitory effect against the dominant fungal strain of moldy peanut kernels. In addition, the effect of the combined EO on the microstructure of mold hyphae was also observed.

## 2. Results

### 2.1. Dominant Fungal Strain from Moldy Peanut Kernels

A dominant fungal strain YQM was isolated from moldy peanuts. Morphological characteristics of strain YQM were shown in Figure 1. On the front side of the solid PDA medium, strain YQM was white at the initial stage, then turned yellow-green at the mature stage (Figure 1A). The colony showed velvet-like protrusions with regular edges, while strain YQM was yellow on the back side (Figure 1B). The phylogenetic tree was constructed based on 18S rRNA gene of strain YQM and closely related reference strains (Figure 2). On the base of point above, the dominant fungal strain YQM was identified as *A*. *flavus*.

### 2.2. Antifungal Activity of Five Single EOs against Strain YQM

The inhibitory effects of five single EOs on the growth of strain YQM are shown in Figure 3. The results revealed that strain YQM could be inhibited by each of EOs (Figure 3A), and the diameter of the inhibition zone was linear with the concentration of EOs (Figure 3B). When the concentration of EOs was 60 µL/mL, the diameters of the inhibition zone of CEO, LEO and CLO were 40.2, 25.0 and 25.2 mm, respectively. The criterion of the inhibition zone experiment is: diameter of inhibition > 20 mm, highly sensitive; diameter of inhibition 10–20 mm, moderately sensitive; and 7 mm < diameter of inhibition < 10 mm, low sensitivity [25]. Strain YQM was highly sensitive to these EOs, among which CEO was the best as a single EO.

### 2.3. Antifungal Activity of Combined EOs against Strain YQM

Based on the results of antifungal activity of five single EOs against strain YQM, three kinds of EOs such as CEO, LEO and CLO were selected for the follow-up experiments due to the relatively better antifungal activity. The three kinds of EOs were mixed in two or three with the same volume, and the final concentration of combined EOs was adjusted to 60 µL/mL. Four kinds of combined EOs such as CLCEO, CEO-CLO, LEO-CLO, and CEO-LEO-CLO were obtained. The antifungal effect of four kinds of combined EOs (1:1 or 1:1:1 mixture) is shown in Figure 4. The cinnamon-litsea combined EO (CLCEO) had the highest antifungal activity on strain YQM (Figure 4A), and its inhibition zone diameter was 40.3 mm (Figure 4B), which was significantly larger than that of other combined EOs (*p* < 0.05). This result clearly elucidated that the combination of CEO and LEO had a better antifungal effect on strain YQM than that of other combination of EOs.

### 2.4. Minimum Inhibitory Concentration (MIC) Value of CLCEO against Strain YQM

The above results revealed that the inhibitory effect of CLCEO on the strain YQM was the best; hence, the minimum inhibitory concentration (MIC) value of CLCEO against strain YQM was determined by the agar dilution method. The combined antifungal effect of CEO and LEO was judged by calculating the fractional inhibitory concentration (FIC) index. The growth of strain YQM treated with different concentrations of CLCEO is summarized in Table 1. The MIC value of CLCEO against strain YQM was 0.0313 µL/mL, which indicated the CLCEO had a good antifungal effect on strain YQM. Based on the criterion [26], the FIC index was 0.28, indicating that the antifungal effect of different kinds of EOs was synergistic. Consequently, it could be concluded that CEO and LEO had a synergistic interaction on inhibiting the growth of strain YQM.

### 2.5. Component of CLCEO with Strongest Antifungal Effect against Strain YQM

The antifungal effect of CLCEO composed of a varied ratios of CEO and LEO was studied. It was noticed that the antifungal effect of CLCEO against strain YQM was the best when the volume ratio of CEO and LEO was 3:5, 4:4 and 5:3, respectively (Figure 5A), while the corresponding inhibition zone diameter was 40.2, 40.3 and 40.0 mm, respectively (Figure 5B). However, there was no significant difference (*p* > 0.05) among the three proportions of CLCEO. In terms of cost, CLCEO (V_CEO_:V_LEO_ = 3:5) was selected as the combined EO with the best antifungal activity and used for further analysis.

### 2.6. Chemical Composition of EOs

The chemical compositions of EOs were determined by GC-MS and analyses are listed in Table 2. A total of 23, 19 and 13 major compounds were identified from CEO, LEO and CLCEO, respectively. CEO was mainly composed of cinnamaldehyde (28.48%), cineole (23.48%) and benzaldehyde (17.20%). In LEO, two main components, citral (31.27%) and limonene (30.56%) were detected. The dominant compounds of CLCEO were aromatic and terpenoid compounds, among which, cinnamaldehyde accounted for 49.33%, followed by citral (34.77%). In addition, eugenol and limonene were also detected in CLCEO, their ratio was 9.11% and 2.83%, respectively.

### 2.7. Effect of EOs on the Microstructure of Strain YQM Mycelia

Scanning electron microscopic (SEM) analysis was adopted to investigate the effect of CEO, LEO and CLCEO (V_CEO_:V_LEO_ = 3:5) on the microstructure of strain YQM mycelia (Figure 6). The microstructure of strain YQM mycelia treated without EOs (the control group) was complete and full, and the thickness was uniform (Figure 6A). In contrast, the mycelia of strain YQM treated with 0.25 µL/mL CEO (Figure 6B) and 2 µL/mL LEO (Figure 6C) was partly deformed and collapsed. Furthermore, the surface of strain YQM mycelia treated with 0.25 µL/mL CLCEO contained large breakages and serious damage such as shrinking, shriveling and sinking occurred (Figure 6D).

## 3. Discussion

As an oil crop rich in protein, lipid and other nutrients, peanuts are easily infected by fungi during the process of field growth or post-harvest storage. Fungal strains will consume the nutrients of peanuts and peanut products. Mycotoxins secreted by fungi could also affect their edible quality, thus posing a serious threat to the health of human beings and animals. Harvested peanuts are susceptible to the infestation of various molds, mainly including *Aspergillus* spp. and *Penicillium* spp., etc. [27,28]. In this study, a dominant fungal strain YQM was isolated from moldy peanuts and identified as *A*. *flavus* by morphological and molecular biological characteristics (Figure 1 and Figure 2). It was reported that *A*. *flavus* could produce B-series aflatoxins such as AFB_1_ and AFB_2_, among which, AFB_1_ was classified as a Group 1 carcinogen by the IARC (1993) because of its involvement in cancer development in humans [29]. Therefore, it is absolutely imperative to find an effective measure to prevent the growth and propagation of *A*. *flavus* strains in peanuts.

Recently, the use of plant-derived EOs to inhibit the growth of fungi and treat various agricultural diseases has received widespread attention [30]. In this study, the antifungal activity of five kinds of EOs such as CEO, LEO, CLO, TEO, and CNO against *A*. *flavus* YQM isolated from moldy peanuts were evaluated. The results showed that all five kinds of EOs exhibited pretty inhibitory effect against strain YQM, among which, CEO had the strongest inhibitory effect on strain YQM (Figure 3). In a similar study, Denkova-Kostova et al. [31] found that cinnamon EO exhibited good antifungal effect on *A*. *flavus* and *A*. *niger*, which might be attributed to cinnamaldehyde, the main antifungal component of cinnamon EO. It was also described that cinnamaldehyde could perform strong antifungal activity against *A*. *niger* and *Fusarium sambucinum* by disru*p*ting the integrity of their cell membrane [32,33]. Achar et al. [34] found that 500–2000 ppm clove EO could inhibit not only the growth of *A*. *flavus* but also the production of aflatoxin. Arasu et al. [35] selected *Acorus calamus* L., *Allium sativum* L., *Mucuna pruriens* (L.) DC., and *Sesamum indicum* L. EOs to constrain the growth of plum fruit spoilage microbes and concluded that *Allium sativum* EO could effectively inhibit the growth of *A*. *flavus* and *A*. *niger* on plum fruit. Sawadogo et al. [36] reported that *Cymbopogon nardus* (L.) EO exhibited high antifungal activity against *A*. *flavus* and *A*. *parasiticus*. Yan et al. [37] determined the antifungal impacts of seven EOs from masson pine, cinnamon, star anise, litsea, camphor, lemon eucalyptus, and camphor on the dominant molds *Trichoderma viride*, *A*. *niger*, and *Penicillium citrinum* inhabited on the surface of bamboos, and found that cinnamon EO, star anise EO, and litsea EO had a good antifungal activity against molds of bamboo, which could reduce molds on the surface of bamboo chips. These researches were similar to the results in this paper, demonstrating the effectiveness of EOs against *A*. *flavus*.

Studies have shown that the antifungal effect of combined EOs is better than that of single EOs by combining two or more EOs to create a synergistic effect [38]. For example, Xiang et al. [26] found that there was a synergistic antifungal effect among cinnamon, oregano, and lemongrass EOs, and the composite EO consisting of three EOs with the volume ratio of 1:5:48 displayed remarkable inhibitory activity against the mycelial growth of *A*. *flavus* and aflatoxin production. In this study, CEO, LEO, and CLO were mixed in different ways to obtain four kinds of combined EOs to explore their antifungal activity on strain YQM. The results indicated that CLCEO exerted the best antifungal effect on strain YQM and the inhibition zone diameter was larger than that of other combination of EOs (Figure 4). The FIC index of CLCEO was calculated according to the MIC (Table 1). The FIC index (FIC = 0.28) showed that the combination of CEO and LEO exhibited synergistic effect against *A*. *flavus* YQM. Similarly, Nikkhah et al. [39] found that the combination of cinnamon and marjoram, thyme and marjoram EOs displayed an additive effect against *Botrytis cinerea*; and the combination of marjoram, cinnamon and thyme exhibited an additive effect against *Penicillium expansum*. Although traditionally used fungicides are effective, the drug resistance and safety issues caused by fungicides have arisen great attention. Aimad et al. [40] compared the antifungal effect of *Origanum compactum* Benth. EO and fluconazole (the traditional fungicide) on *A*. *flavus* and the results showed that the MIC value of *Origanum compactum* Benth. EO was 6.25 µg/mL, lower than that of fluconazole (256 µg/mL). Studies have also shown that EO can act as enhancers of commercial antifungal agents to enhance the antifungal activity, and EOs are potential alternatives to synthetic fungicides [41,42,43]. In addition, the optimal volume ratio of CLCEO with strongest antifungal effect against *A*. *flavus* YQM was also determined in this research. It was found that CLCEO composed of CEO and LEO with a volume ratio of 3:5, 4:4 and 5:3 (*v/v* ) showed the best antifungal effect against *A*. *flavus* YQM (Figure 5). Finally, CLCEO consisting of CEO and LEO with the volume ratio of 3:5 was chosen to be the best due to the costs of CEO and LEO.

Studies have reported that the nature bioactive compounds in EOs mainly contribute to the antifungal properties [40,44]. In this study, the chemical compositions of CEO, LEO and CLCEO were analyzed by GC-MS (Table 2). According to the results, cinnamaldehyde (28.48%), cineole (23.48%) and benzaldehyde (17.20%) were the main components of the CEO, which was consistent with that of previous reports. Xiang et al. [26] found that cinnamaldehyde was the main chemical component of cinnamon EO. Hlebová et al. [45] also reported that cinnamaldehyde was the most abundant component in cinnamon EO. For LEO, citral (31.27%) and limonene (30.56%) were the main components, which was consistent with that of Zhao et al. [46] They analyzed litsea EO by GC-MS and found that the important components included α-pinene, limonene, linalool, citral, etc., among which, the predominant constituents were citral and limonene. In addition, the main components of CLCEO were also detected, and the results showed that cinnamaldehyde (49.33%) and citral (34.77%) accounted for the two highest proportions. The compositions of EOs used in different studies vary greatly, which may be related with the extraction methods, but the main components of EOs were the same. Liang et al. [47] compared the effects of subcritical n-butane and ethanol extraction on the composition of cinnamon EO and found that cinnamaldehyde was the major compound in both extraction methods.

In general, the most common conclusion about the antifungal mechanisms was the damage to the cell membrane and cell wall [48]. In this research, the potential antifungal mechanism of EOs against *A*. *flavus* YQM was preliminarily investigated. The effect of CEO, LEO and CLCEO on the microstructure of *A*. *flavus* YQM mycelia was analyzed by SEM. The hyphae of the LEO treatment group were less shriveled than that of the CEO treatment group, which was consistent with the results of the previous inhibition zone experiment. Compared with single EOs, combined EO treatment caused more serious damage to mycelia (Figure 6). Based on the results, it was speculated that the possible antifungal mechanism of CLCEO on *A*. *flavus* YQM may be that CLCEO damaged the integrity of the cell wall and cell membrane, resulting in the leakage of cellular contents and causing cell apoptosis, which might be consistent with the results of previous research [49,50]. For example, Wang et al. [51] explored the antifungal properties and potential mechanisms of clove EO against *Colletotrichum gloeosporioides* isolated from sweet cherry, and the SEM results showed that the morphology and ultrastructure of *Collelotrichum gloeosporioides* treated with clove EO were destroyed and altered. Moreover, Sarathambal et al. [52] studied the inhibitory effect of *Pimenta dioica* (L.) leaf EO on *A*. *flavus*, and several irreversible morphological changes such as loss of pigmentation and irregular maturity of conidiophores were observed under a light microscopy. In addition, Brandão et al. [53] evaluated the antifungal and anti-mycotoxigenic effects of *Eremanthus erythropappus* (DC.) MacLeish EO against *A*. *carbonarius*, *A*. *flavus*, and *A*. *ochraceus*. The results of SEM showed that the integrity of the mycelium was destroyed and the conidial head was deformed or not formed after treatment with the *Eremanthus erythropappus* EO.

The U.S. FDA (Food and drug administration) has classified cinnamon, citronella, thyme, and other EOs as GRAS (Generally Recognized as Safe) status. The safety of EOs is related with various factors, such as the dosage, action time and receptor [54]. In general, small amount of EOs is safe for most people [55]. Research has proven that there was no obvious toxicity in mice after oral administration of thymus vulgaris [56]. In another study, there were no safety concerns by adding litsea berry EO into the target animal feed [57]. In recent years, EOs have also been widely used as new coatings or fungicides alternatives in the preservation of fruits, vegetables and other foods [58,59,60]. Therefore, there is great chance for EOs to be used as anti-mildew agents for peanut storage. Although research has made some progress, the specific antifungal mechanisms of CLCEO are not fully understood. Additionally, the effect of CLCEO on the quality of peanut during storage process has not been investigated. Further studies should focus on the specific antifungal mechanism of CLCEO on *A*. *flavus* YQM and the effect of CLCEO on the quality of stored peanuts.

## 4. Materials and Methods

### 4.1. Materials

The peanut kernels (Yuanza 9102) were harvested in September, 2020, provided by farmers from Zhengyang city, Henan province, China, and the initial moisture content was 5% (*w*/*w*). Five kinds of EOs such as CEO, LEO, CLO, TEO, and CNO used in this study were purchased from Zhongxiang Botanical Co., Ltd. (Ji’an, Jiangxi, China) and stored at 4 °C for further use. Potato dextrose agar (PDA) was bought from Beijing Aobox Biotechnology Co., Ltd. (Beijing, China); potato dextrose broth (PDB) was acquired from Qingdao Haibo Company (Qingdao, Shandong, China); Tween-20 was obtained from Beijing Solarbio Technology Co., Ltd. (Beijing, China) and kept under room temperature for later use. All reagents used in this study were analytical-grade.

### 4.2. Isolation and Identification of the Dominant Fungal Strain from Moldy Peanut Kernels

The moisture content of peanut kernels was adjusted from 5% to 9.7%. Around 500 g of peanut kernel samples with 9.7% moisture were put into a sterile transparent plastic bag and stored in an HWS humidity chamber (Ningbo Southeast Instrument Co., Ltd., Ningbo, Zhejiang, China) at 30 °C until the samples turned moldy. Then 25 g of moldy peanut kernels were mixed with 225 mL sterile water, and vortexed fully in MVS-1 Vortex Mixer (Jinbeide Industry and Trade Co., Ltd., Beijing, China). Subsequently, the suspension was spread onto PDA plates and the plates were incubated at 28 °C for 3 days. After three times’ streaking, single colonies were obtained, among which, strain YQM was picked out for further experiments.

Strain YQM was inoculated on PDA plate and incubated at 28 °C for 5 d. Then the morphology of strain YQM was observed under UB203i Upright Biological Microscope (Chongqing Uop Optoelectronics Technology Co., Ltd., Chongqing, China). Morphological characteristics of strain YQM were compared with the descriptions of GB/T 4789.16-2016. Genomic DNA of strain YQM was extracted with TIAN quick FFPE DNA Kit (Tiangen Biochemical Technology Co., Ltd., Beijing, China) and the universal fungal primer set (ITS1: 5′-TCCGTAGGTGAACCTGCGG-3′ and ITS4: 5′-TCCTCCGCTTATTGATATGC-3′) was used to amplify 18S rRNA gene fragment. The 18S rRNA gene of strain YQM was sequenced by BEIJING LIUHE Co., Ltd (Beijing, China). The 18S rRNA gene sequence was compared by using the database of National Center for Biotechnology and Information Basic Local Alignment Search Tool (NCBI BLAST, National Center for Biotechnology and Information Basic Local Alignment Search Tool. Available online: https://blast.ncbi.nlm.nih.gov/Blast.cgi, accessed on 6 March 2022). The phylogenetic tree based on 18S rRNA gene sequences was constructed by MEGA X software [61]. In addition, the 18S rRNA gene sequence of strain YQM was deposited to the NCBI GeneBank (National Center for Biotechnology and Information. Available online: https://www.ncbi.nlm.nih.gov/, accessed on 6 March 2022), and the accession number was MT903389.

### 4.3. Determination of the Antifungal Activity of EOs against Strain YQM

The antifungal activity of EOs against strain YQM was evaluated by the inhibition zone experiment described by Xia et al. [25] with several modifications. In brief, EOs were dissolved in 1% sterile Tween-20 solution to obtain EOs solutions with the concentration of 10, 20, 30, 40, 50, and 60 µL/mL, respectively. The preparation of strain YQM suspension was based on the method of Ma et al. [62]. Firstly, the spores of strain YQM incubated at 28 °C for 5 d were collected. Secondly, the concentration of fungal spores was determined by using a hemocytometer. Then the final concentration of strain YQM spores was adjusted to 5 × 10^5^–6 × 10^5^ spores/mL. The strain YQM spore suspension (100 µl) was sucked and spread evenly onto the PDA plate. Next, a circular hole with a diameter of 7 mm was punched in the center of the plate by a puncher, and 100 µL of different kinds of EOs solution with the concentration of 10, 20, 30, 40, 50, and 60 µL/mL was added to the hole, respectively. Subsequently, the plates were placed at room temperature (around 25 °C) for one hour to make the EOs fully diffuse and incubated at 28 °C for 3 days. Finally, the diameter of the inhibition zone was measured by the cross method and the pictures were taken for further use. The same amount of 1% Tween-20 solution was used to replace EOs in the control group and each experiment was done in triplicate. Similarly, the inhibition zone experiment was adopted to evaluate the antifungal effect of combined EOs on strain YQM.

### 4.4. Determination of MIC Value of CLCEO against Strain YQM

The MIC value of CLCEO was determined by the agar dilution method with slight modifications [63]. Strain YQM spore suspension (100 µL) was evenly spread on sterile PDA plates containing 0.0313, 0.0625, 0.125, 0.25, 0.5, 1, and 2 µL/mL CLCEO. Then the plates were incubated at 28 °C for 4 days. The lowest concentration of CLCEO that did not allow the visible growth of colonies was regarded as the MIC value. In addition, fractional inhibitory concentration (FIC) was used to determine the combined antifungal effect of CEO (A) and LEO (B) [39].
(1)$${\mathrm{FIC}}_{A}=\frac{{\mathrm{MIC}}_{A}\;\mathrm{combination}}{{\mathrm{MIC}}_{A}\;\sin\mathrm{gle}}{\;\mathrm{FIC}}_{B}=\frac{{\mathrm{MIC}}_{B}\;\mathrm{combination}}{{\mathrm{MIC}}_{B}\;\sin\mathrm{gle}}\;{\mathrm{FIC}}_{\mathrm{index}}={\mathrm{FIC}}_{A}+{\mathrm{FIC}}_{B}$$

The results were explained as follows: the synergistic effect (FIC ≤ 0.5), the additive effect (0.5 < FIC ≤ 1.0), the indifferent effect (1 < FIC ≤ 2) and the antagonistic effect (FIC > 2) [26].

### 4.5. Determination of the Component of CLCEO with Strongest Antifungal Effect against Strain YQM

In order to obtain CLCEO with the best antifungal effect on strain YQM, CEO, and LEO were mixed with the volume proportions of 1:7, 2:6, 3:5, 4:4, 5:3, 6:2, and 7:1, respectively. The final concentration of eight kinds of CLCEO was all adjusted to 60 µL/mL. The inhibition zone test was performed to evaluate the antifungal activity of CLCEOs on strain YQM. Eight kinds of CLCEO solution (100 µL) were added to the circular hole, respectively. The plates were placed at room temperature (25 °C) for 1 h and then incubated at 28 °C for 3 days.

### 4.6. Determination of the Chemical Composition of EOs

The chemical composition of CEO, LEO and CLCEO were analyzed by gas chromatograph-mass spectrometry (GC-MS) and according to the method of Wang et al. [64]. The samples were processed as follows: 100 mg EO was added into 10 mL n-hexane and vortexed for 5 min to fully dissolve evenly. After being filtered with a 0.22 μm filter membrane, the supernatant (2 μL) was subjected to GC-MS analysis.

### 4.7. Determination of the Effect of EOs on the Microstructure of Strain YQM Mycelia

The mycelium structure observation was conducted according to the descriptions of Wang et al. [51] and Achar et al. [34] with slight modifications. Strain YQM was cultured in PDB medium and shaken at the speed of 150 r/min at 28 °C for 2 days, then CEO, LEO, and CLCEO dissolved by 1% Tween-20 were added to the PDB medium to achieve final concentrations of 0.25 µL/mL, 2 µL/mL, and 0.25 µL/mL, respectively. The mycelia treated without EOs were used as the control group. After culturing for one day, the mycelia of strain YQM were collected. The mycelia were rinsed with sterile PBS for three times, fixed with 2.5% (*w*/*w*) glutaraldehyde solution, and stored in refrigerator at 4 °C overnight. The mycelia were rinsed with sterile PBS for three times (10 min per time) again, and then rinsed with sterile water for two times. The mycelia were dehydrated with 30%, 50%, 70%, and 90% (*v*/*v*) ethanol for 10 min, respectively, and anhydrous ethanol for 15 min. Next, the mycelia were dehydrated with the tert-butanol-ethanol solution with the volume ratio of 1:1 and 2:1 for 10 min, respectively. Finally, after washing twice with tert-butanol solution, the tert-butanol solution containing mycelia was obtained. The mycelia samples were pre-frozen at −80 °C for 2 h and then frozen-dried. After drying, the samples were sprayed with gold, observed and photographed under Quanta 250 FEG Scanning Electron Microscope (FEI Company, Hillsboro, OR, USA).

### 4.8. Statistical Analysis

All experiments in this study were performed in triplicate and the data were presented as mean ± standard deviation (SD). Data were analyzed using SPSS (20.0) software and Microsoft Excel (2016). Duncan’s method was used to evaluate the significant differences between the data, and statistically significant differences were set at *p* < 0.05. Some figures shown were drawn using Origin Lab (2018) software.

## 5. Conclusions

In this study, a dominant fungal strain YQM was isolated from moldy peanut kernels and identified as *A*. *flavus*. Five single EOs such as CEO, LEO, CLO, TEO, and CNO had antifungal effects on *A*. *flavus* YQM, and as the concentration of EOs increased, the inhibitory effects also increased. Furthermore, CEO and LEO showed synergistic effect on restraining the growth of *A*. *flavus* YQM, and the ratio of CLCEO with strongest antifungal effect against *A*. *flavus* YQM was V_CEO_:V_LEO_ = 3:5. GC-MS results showed that the major components of CLCEO were cinnamaldehyde and citral. According to the results of SEM, it was preliminarily speculated that CLCEO might inhibit the growth of *A*. *flavus* YQM by destroying the mycelial structure. In conclusion, CLCEO has the potential to be used as an efficient anti-mildewing agent to improve the shelf life of peanuts during the storage process.

## Figures and Tables

**Figure 1 foods-11-01586-f001:**
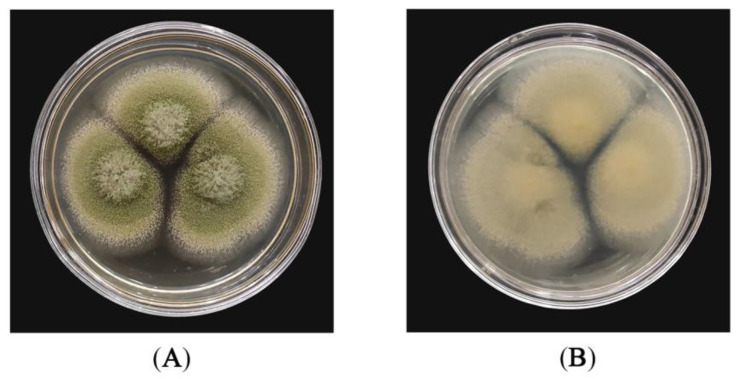
Morphological characteristics of strain YQM: (**A**) front side; (**B**) back side.

**Figure 2 foods-11-01586-f002:**
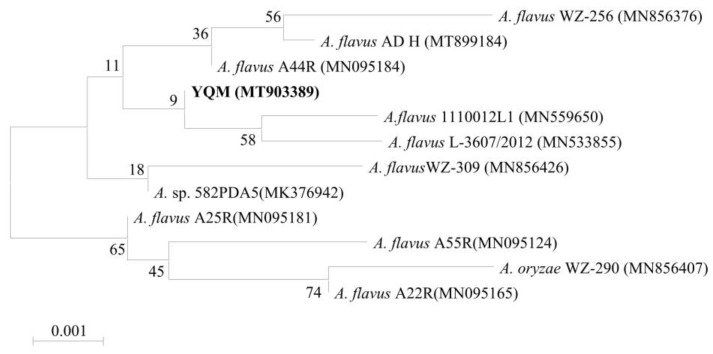
Phylogenetic tree of strain YQM and closely related reference strains of *Aspergillus flavus* based on 18S rRNA gene.

**Figure 3 foods-11-01586-f003:**
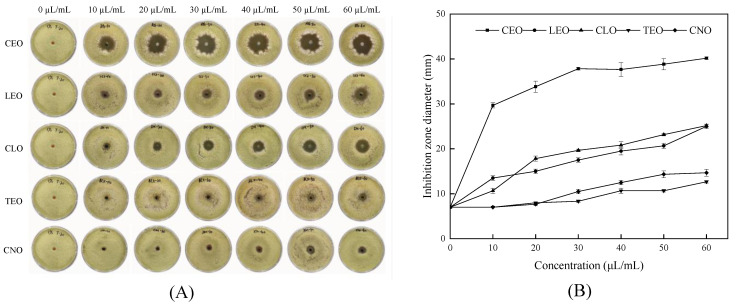
Antifungal effect of different concentrations of five single essential oil (EOs) on strain YQM (CEO, cinnamon EO; LEO, litsea EO; CLO, clove EO; TEO, thyme EO; and CNO, citronella EO): (**A**) growth of strain YQM on 9-cm plate media; (**B**) quantification of inhibition as zone diameter. Values are average values ± standard deviation (biological replicates, n = 3).

**Figure 4 foods-11-01586-f004:**
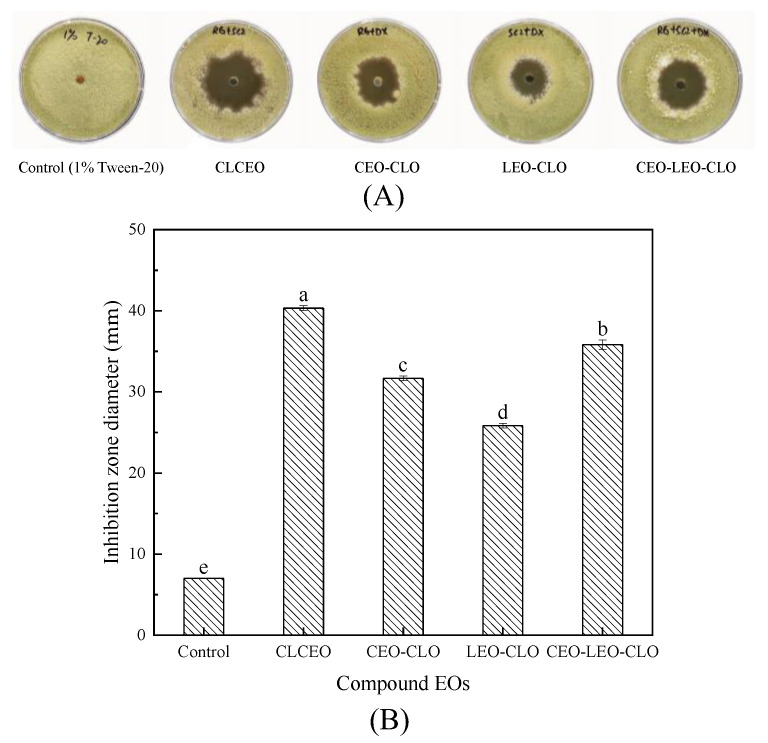
Antifungal effect of four combined EOs on strain YQM (CLCEO: cinnamon-litsea combined EO): (**A**) growth of strain YQM on 9-cm plate media; (**B**) quantification of inhibition as zone diameter. Values are average values ± standard deviation (biological replicates, n = 3). Values with different superscript lowercase letters show significant differences among treatments (*p* < 0.05).

**Figure 5 foods-11-01586-f005:**
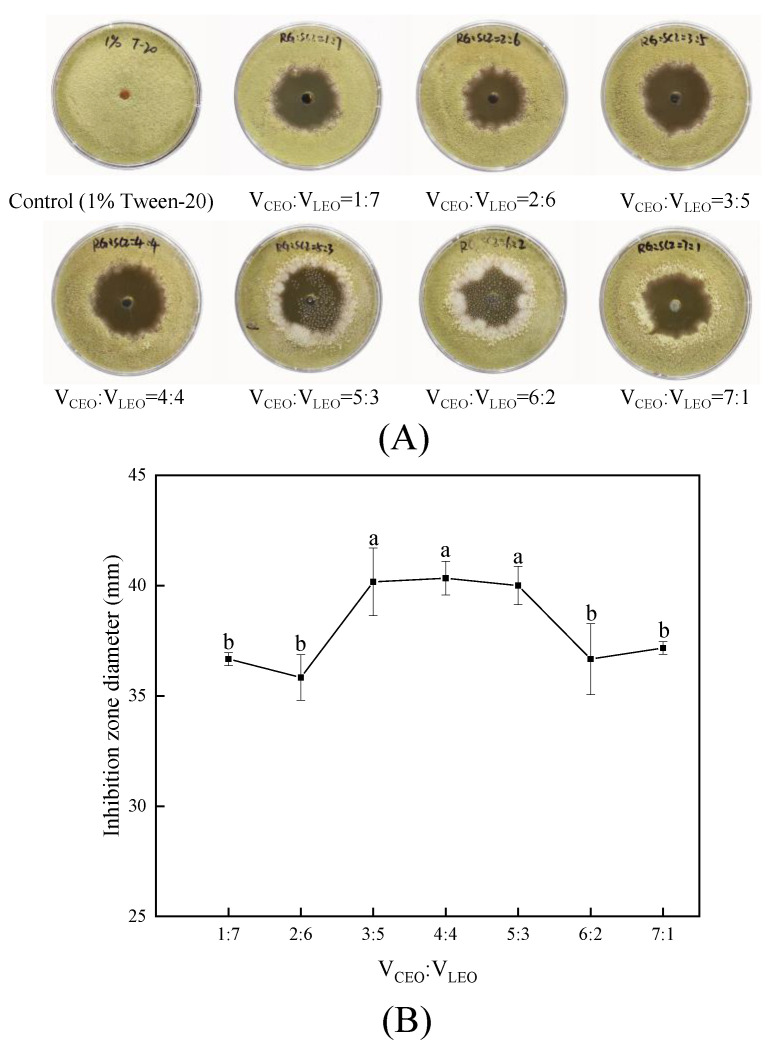
Antifungal effect of different combinations of CLCEO on strain YQM: (**A**) growth of strain YQM on 9-cm plate media; (**B**) quantification of inhibition as zone diameter. Values are average values ± standard deviation (biological replicates, n = 3). Values with different superscript lowercase letters show significant differences among treatments (*p* < 0.05).

**Figure 6 foods-11-01586-f006:**
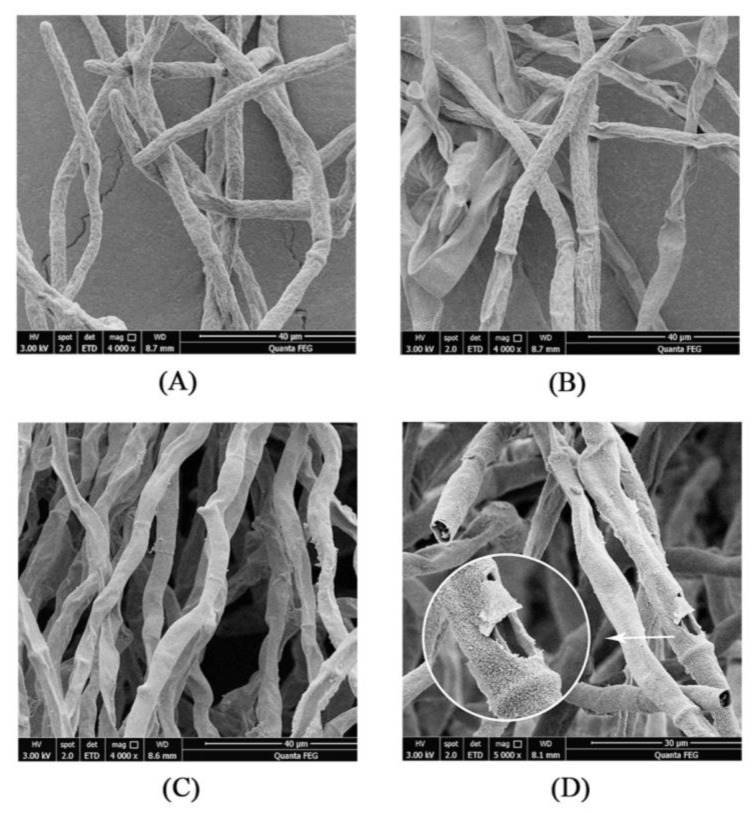
Effects of EOs on the microstructure of strain YQM mycelium: (**A**) control; (**B**) treated with CEO at 0.25 µL/mL; (**C**) treated with LEO at 2 µL/mL; (**D**) treated with CLCEO at 0.25 µL/mL.

**Table 1 foods-11-01586-t001:** Growth of strain YQM treated with different concentrations of CLCEO.

EOs	Different Concentrations of EOs									MIC
	2	1	0.5	0.25	0.125	0.0625	0.0313	0.0156	Blank	(µL/mL)
CEO	—	−	−	−	−	−	+	+	+	0.0625
LEO	−	−	—	w	w	+	+	+	+	0.5
CLCEO	−	−	−	−	−	−	−	+	+	0.0313

Note: Strain YQM growth was evaluated as +, growth; −, no growth; 1% Tween-20 solution was used for blank; MIC: minimum inhibitory concentration.

**Table 2 foods-11-01586-t002:** Chemical compounds of individual and combined essential oils.

Compounds	Retention Time/min	CEO/%	LEO/%	CLCEO/%
2-Methylheptane	5.517	0.31	—	−
n-Octane	6.181	1.72	−	−
2,4-Dimethyl-heptane	6.697	1.43	−	−
4-Methyloctane	7.846	0.19	−	−
Cyclofenchene	10.065	0.18	−	−
Pinene	10.537	1.53	4.47	0.59
Camphene	11.215	1.04	1.07	0.10
Benzaldehyde	11.945	17.20	−	−
Sabinene	12.260	−	9.43	0.46
*trans*-pinocarveol	12.432	−	3.90	−
Myrcene	13.028	−	2.96	0.21
Decane	13.437	1.29	−	−
α-Terpinene	14.423	−	0.13	−
4-Methyldecane	14.594	0.31	−	−
Cineole	15.254	23.48	−	−
limonene	15.318	−	30.56	2.83
γ-Terpinene	16.760	−	0.34	−
Sabinene hydrate	17.542	−	0.37	−
Isogeranialdehyde	17.75	−	3.87	0.76
Linalool	19.492	−	3.13	0.21
Phenethyl alcohol	20.844	4.75	−	−
7-methyl-3-methylene-6-octena	21.974	−	1.79	−
Citronellal	22.535	0.54	1.14	0.19
Dodecane	25.287	3.76	−	−
α-Terpineol	25.592	−	0.77	−
Cinnamaldehyde	27.202	28.48	−	49.33
Citral	28.266	0.35	31.27	34.77
1,3-Di-tert-butylbenzene	28.849	0.99	−	−
2,4-Dimethyldodecane	29.346	0.51	−	−
4,6-Dimethyldodecane	30.303	1.01	−	−
γ-Elemene	34.300	−	0.13	−
2-Methyl-3-phenyl-2-propenal	34.605	0.98	−	−
α-Terpineyl Acetate	35.247	−	0.22	−
Piperitene	36.839	−	0.37	−
2,6-di-tert-butyl-4-methylphenol	37.11	−	−	0.87
Caryophyllene	39.453	3.52	2.53	0.56
Cinnamyl ester	41.123	2.58	−	−
Eugenol	47.633	3.55	−	9.11
Total	−	99.71	98.42	99.99

Note: −, not detected; CEO, cinnamon essential oil; LEO, litsea essential oil; CLCEO: cinnamon-litsea combined EO.

## Data Availability

The data used in this study are available in this article.

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
