# Peer review of "The Antifungal Activity of Cinnamon-Litsea Combined Essential Oil against Dominant Fungal Strains of Moldy Peanut Kernels"

_foods, 2022, doi:10.3390/foods11111586_

Round 1
Reviewer 1 Report
The authors should elaborate more on their findings compared to other studies, to their importance, as well as to safety issues
Please include some latest research findings, updated reviews (2022) in introduction and
discussion part related to the topic.
Make corrections in typographical errors as much as possible
English should be improved; grammar need for enhancement in many sentences and
paragraphs.
Increase figure size and enhancement resolution
Author Response
1. The authors should elaborate more on their findings compared to other studies, to their importance, as well as to safety issues.
A: Thanks for the valuable advice. This point has been modified in the manuscript and the safety issues have been added in the sixth paragraph of section 3.
2. Please include some latest research findings, updated reviews (2022) in introduction and discussion part related to the topic.
A: Thanks for the valuable advice. The updated reviews (2022) on essential oils against fungi have been added in the Introduction and Discussion section (Reference [2], [4], [9], [13], [14], [18], [36], [40], [44], [48] in the revised manuscript).
3. Make corrections in typographical errors as much as possible. English should be improved; grammar need for enhancement in many sentences and paragraphs
A:Thanks for the valuable advice. The typographical errors have been corrected and English has been improved in the revised manuscript.
4. Increase figure size and enhancement resolution.
A: Thanks for the valuable advice. The size of figures has been increased and all figures with high resolution (minimum 1000 pixels width/height, or a resolution of 300 dpi) have been used in the revised manuscript.
Reviewer 2 Report
The manuscript entitled The antifungal activity of cinnamon-litsea combined essential oils against dominant fungal strains of moldy peanut kernels is well-structured, comprehensive and the literature is current and appropriate. All used experiments are chosen right and the presented results are more than interesting. In my opinion this paper should be accepted after the minor revision. My main concerns are related to:
Section Materials and methods:
- Please, explain why the Tween20 solution is used for preparation of EO dilutions and not DMSO;
- Also, describe the reason for choosing these EO in the study
- If the diameter of the hole in the center of culture media is 7 mm, is it correct that the results of the inhibition zone diameter between 7 and 10 mm should be considered as low antifungal activity?
Conclusion section:
- How this effective ratio of the tested EOs could be practically used in prolonging the shelf-life of peanuts?
Author Response
1. Section Materials and methods: Please, explain why the Tween20 solution is used for preparation of EO dilutions and not DMSO
Thanks for the valuable question. At the beginning of the experiments, it was found that DMSO interfered with the experimental results of the inhibition zone test. Therefore, 1% Tween 20 was chosen to prepare the EO dilutions.
2. Also, describe the reason for choosing these EO in the study
Thanks for the valuable question. Previous studies have shown that cinnamon, litsea, clove, thyme and citronella essential oils could inhibit the growth of bacteria and fungi, and they are widely used in the fields of food and medicine. Therefore, we tried to develop an efficient EO antimildew agent to control the growth of Aspergillus flavus in peanut storage.
3. If the diameter of the hole in the center of culture media is 7 mm, is it correct that the results of the inhibition zone diameter between 7 and 10 mm should be considered as low antifungal activity?
Thanks for the valuable advice. The criterion of low sensitivity has been corrected as: 7 mm < the inhibition zone diameter <10 mm in the revised manuscript (Line 116-117).
4. Conclusion section: How this effective ratio of the tested EOs could be practically used in prolonging the shelf-life of peanuts?
Thanks for the valuable question. In next experiments, the combined essential oil will be microencapsulated and the combined EO microcapsules will be used for peanut storage and the effects of the microcapsules on microorganisms and peanut quality during postharvest storage will be investigated.
Reviewer 3 Report
The authors of the manuscript “The antifungal activity of cinnamon-litsea combined essential oil against dominant fungal strains of moldy peanut kernels” present an interesting work about the antifungal activity of 5 different essential oils (EOs) and several combination of them. Authors use different tools to show the antifungal activity, for example inhibition halos of growth at different concentrations, MIC and SEM analysis. The manuscript is clearly written and the results are of interest but, it contains some flawn to be improved before acceptance.
Line 32: please delete fullstop after Aspergillus.
Line 98-100: Please remove this sentence. The kind of growth of the fungus in PDA plates depends on many factors (temperature, supplier of the PDA medium, sterilization cycle…). 18S rRNA its ok for classification.
Line 101 – 104: Please arrange differently the sentences (on the base of point above).
Figure 2: please, change the font size. A smaller one on phylogenetic tree helps the figure.
Line 114: Authors write a concentration (60 µL/mL) without explaining what is the milliliter. Is it the solvent of essential oil? a brief recall of matmeth chapter is needed.
Line 135: authors wrote “This result clearly elucidated that the combination of CEO and LEO had a better antifungal effect on strain YQM than that of single CEO or LEO”. At the end, the inhibition halos of CEO and CLCEO are 40,2 mm and 40,3 mm respectively, they have the same MIC and it is available only a CLCEO composition by GC-MS. Authors should put CEO halo in the same sistem of figure 4B and check if it is significantly larger.
Line 169: Every kind of discussion need to know at leat two essential oil composition (CLCEO plus CEO or LEO). In this case it is better to show in the table also literature data cited in discussion chapter.
Line 182: please, change µL/L in µL/mL (I think).
Figures 3A, 4A, 5A: All the experiments with inihibition of growth need a positive control with a commonly used fungicide. It is not necessary a new experiment, the data can be retrievied fom bibliography (probably). This will permit a wide range discussion, for example: if 1 mg of fungicide is equal to 2 mL of essential oil, does synthetic fungicide still dangereous and/or economic than essential oil?
Discussion: This chapter need at leat 1 paragraph about essential oil toxicity on animal (or mammalian) cells. Basically essential oil could enter in food chain.
Author Response
1. Line 32: please delete fullstop after Aspergillus
Thanks for the valuable advice. The fullstop after Aspergillus has been deleted in the revised manuscript (Line 31).
2. Line 98-100: Please remove this sentence. The kind of growth of the fungus in PDA plates depends on many factors (temperature, supplier of the PDA medium, sterilization cycle…). 18S rRNA its ok for classification.
Thanks for the valuable advice. This sentence has been deleted in the revised manuscript.
3. Line 101 – 104: Please arrange differently the sentences (on the base of point above).
Thanks for the valuable advice. This sentence has been rearranged on the basis of the above point in the revised manuscript (Line 100-103).
4. Figure 2: please, change the font size. A smaller one on phylogenetic tree helps the figure.
Thanks for the valuable advice. The font size has been reduced in the revised manuscript.
5. Line 114: Authors write a concentration (60 µL/mL) without explaining what is the milliliter. Is it the solvent of essential oil? a brief recall of matmeth chapter is needed.
Thanks for the valuable question. The concentration of 60 µL/mL means that 60 µL EO is dissolved into 1 mL 1% Tween20 solvent. The preparation of different concentrations of EO solutions has been described in detail in the subsection 4.3 of the revised manuscript (Line 351-352).
6. Line 135: authors wrote “This result clearly elucidated that the combination of CEO and LEO had a better antifungal effect on strain YQM than that of single CEO or LEO”. At the end, the inhibition halos of CEO and CLCEO are 40,2 mm and 40,3 mm respectively, they have the same MIC and it is available only a CLCEO composition by GC-MS. Authors should put CEO halo in the same system of figure 4B and check if it is significantly larger.
Thanks for the valuable questions. The sentence “This result clearly elucidated that the combination of CEO and LEO had a better antifungal effect on strain YQM than that of single CEO or LEO” has been replaced by “This result clearly elucidated that the combination of CEO and LEO had a better antifungal effect on strain YQM than that of other combination of EOs” in the revised manuscript.
Experiments of minimum inhibitory concentration (MIC) value of CLCEO against strain YQM were repeated and the new results showed that 0.0313 µL/mL CLCEO can effectively inhibit the growth of strain YQM, which was indicated that the MIC value of CLCEO against strain YQM was 0.0313 µL/mL. The new data has been updated in the revised manuscript.
In order to compare the chemical compositions of single EOs and CLCEO, the composition of CEO and LEO was supplemented in the revised manuscript.
The diameters of inhibition zone of single CEO and CLCEO were analyzed, and the results showed that there were no significant differences between the diameters. However, the data of single CEO was unsuitable for being added into Figure 4, which may mislead the readers that the investigation of combining single CEO and LEO was not meaningful. Though the inhibition diameter was not significantly different, the MIC value of CLCEO was half of that of single CEO, which indicated that lower concentration of CLCEO can achieve the same antifungal effect of double amount of CEO. Obviously, compared with single CEO, the use of CLCEO will reduce the costs and usage.
7. Line 169: Every kind of discussion need to know at least two essential oil composition (CLCEO plus CEO or LEO). In this case it is better to show in the table also literature data cited in discussion chapter.
Thanks for the valuable advice. The chemical composition of CEO and LEO was added in Table 2 in the revised manuscript. The new results and discussion about CEO and LEO have also been added in the subsection 2.6 and Discussion section (In the fourth paragraph of section 3). The materials and methods of subsection 4.6 have also been modified accordingly (Line 389).
8. Line 182: please, change µL/L in µL/mL (I think).
Thanks for the valuable advice. The mistake has been corrected in the revised manuscript (Line 184).
9. Figures 3A, 4A, 5A: All the experiments with inhibition of growth need a positive control with a commonly used fungicide. It is not necessary a new experiment, the data can be retrieved from bibliography (probably). This will permit a wide range discussion, for example: if 1 mg of fungicide is equal to 2 mL of essential oil, does synthetic fungicide still dangerous and/or economic than essential oil?
Thanks for the valuable advice. The data and discussion about commonly used fungicide has been added in the revised manuscript (Line 245-248).
10. Discussion: This chapter need at least 1 paragraph about essential oil toxicity on animal (or mammalian) cells. Basically essential oil could enter in food chain.
Thanks for the valuable advice. The safety of EOs has been discussed in the revised manuscript (In the newly added sixth paragraph of section 3).

Round 2
Reviewer 3 Report
The authors fully answered to the previous remarks, the manuscript can be published.